# Occurrence of Multidrug Resistant *Escherichia coli* in Raw Meat and Cloaca Swabs in Poultry Processed in Slaughter Slabs in Dar es Salaam, Tanzania

**DOI:** 10.3390/antibiotics10040343

**Published:** 2021-03-24

**Authors:** Fauster X. Mgaya, Mecky I. Matee, Amandus P. Muhairwa, Abubakar S. Hoza

**Affiliations:** 1Department of Veterinary Microbiology and Parasitology, Sokoine University of Agriculture, Morogoro 67000, Tanzania; abhoza@sua.ac.tz; 2Department of Microbiology and Immunology, Muhimbili University of Health and Allied Sciences, Dar es Salaam 65001, Tanzania; mmatee@muhas.ac.tz; 3Department of Veterinary Medicine and Public Health, Sokoine University of Agriculture, Morogoro 67000, Tanzania; apm@sua.ac.tz

**Keywords:** multidrug resistant, *Escherichia coli*, chicken meat, cloaca, Dar es Salaam, Tanzania

## Abstract

This cross-sectional study was conducted between January and June 2020, in five large poultry slaughter slabs in Dar es Salaam, Tanzania. Purposive sampling was used to select broilers and spent layers, from which meat and cloaca swabs were collected to determine the occurrence of multidrug resistant (MDR) *Escherichia coli*. Identification of isolates was done using API 20E, and antimicrobial susceptibility testing was performed as per CLSI (2018) guidelines. EBSL (*CTX-M*, TEM, SHV) and plasmid mediated quinolone (*qnrA*, *qnrB*, *qnrS* and *aac(6′)-Ib-cr*) were screened using PCR. Out of 384 samples, 212 (55.2%) were positive for *E. coli*, of which 147 (69.3%) were resistant to multiple drugs (MDR). Highest resistance was detected to tetracycline (91.9%), followed by sulfamethoxazole-trimethoprim (80.5%), ampicillin (70.9%), ciprofloxacin (40.2%) and 25% cefotaxime, gentamycin (10.8%) and imipenem (8.6%) (95% CI, *p* < 0.01). Out of the *E. coli*-positive samples, ten (10/212) (4.7%) were ESBL producing *E. coli,* of which CTX-M was detected in two isolates and quinolones resistant gene (*qnrS)* in eight, while TEM, SHV, *qnrA*, *qnrB* and *aac(6′)-lb-cr* were not detected. The high level of resistance and multidrug resistance imply these antibiotics are ineffective, add unnecessary cost to poultry farmers and certainly facilitate emergence and spread of resistance.

## 1. Introduction

In Tanzania, the demand for chicken meat was projected to increase from 130,000 tons in 2017 to 465,600 tons in 2020 [1], largely due to an increase in urbanization and trade liberation of live animals and products [2]. Dar es Salaam, which is the commercial city of the country, with an estimated population of 4,364,541 people, is by far the largest consumer of poultry meat in Tanzania [3].

Poultry farming in Dar es Salaam is done both in urban and peri-urban areas, often in overcrowded and unhygienic conditions [4]. Such conditions are frequently associated with occurrence of diseases and use of excessive antimicrobials. Several studies conducted in Tanzania have shown both overuse of antibiotics and high levels of resistant organisms in poultry production systems [2,5,6]. Antibiotics are commonly used for disease prevention and therapeutic purposes in poultry production systems, are obtained over the counter and are administered without advice of veterinary officers [7]. The knowledge of most poultry keepers on prudent use of antibiotics and their effect is low, and antimicrobial prescribers and unregistered veterinary drug dealers also have little prescription knowledge, which all together create an environment for emergence and spread of antimicrobial resistance [5]. Metaphylaxis is also very common among poultry keepers [8], exposing even healthy chickens to unnecessary antimicrobials.

We conducted this study in Dar es Salaam, where the demand for poultry meat and products is the highest in the country and the use of antimicrobials among poultry keepers is very high [5,9]. We determined the occurrence of multidrug-resistant *E. coli* in raw chicken meat and in cloaca as well as the occurrence of extended spectrum beta lactamase, specifically *CTX-M*, TEM and SHV, and plasmid mediated quinolone-resistant genes (*qnrA*, *qnrB*, *qnrS* and *aac(6′)-Ib-cr*).

## 2. Results

### 2.1. Prevalence of E. coli in Raw Chicken Meat and Cloaca in Broiler and Spent Layers

A total of 384 chicken meat and cloaca swabs samples were collected in the five selected poultry slabs in Dar es Salaam. Out of these samples, 212 (55.2%) were positive for *E. coli.* Of the isolated strains, 147 (69.3%) were resistant to more than three tested antibiotics of different classes. The slab with the highest proportion of MDR isolates was at Stereo in the Temeke District (18/19, 94.7%), followed by Shekilango in the Ubungo district (37/43, 86%), Manzese in the Ubungo district (28/40, 70%), Mtambani in the Kinondoni district (14/20, 70%) and Kisutu in the Ilala District (50/90, 55.6%) (Table 1).

### 2.2. Antibiotic Resistance Rates in E. coli Isolates

Overall, the highest resistance was detected in tetracycline (91.9%), followed by trimethoprim-sulfamethoxazole (80.5%), ampicillin (70.9%), ciprofloxacin (40.2%), cefotaxime (22.5%), 10.8% gentamycin (10.8%) and imipenem (3.3%) (Figure 1).

Of the 147 MDR *E. coli* isolates, 49% showed resistance to at least three classes of antibiotics, 33.3% to at least four classes, 14.3% resistant to five classes, 2.7% resistant to six classes, and one isolate (0.7%) was resistant to all seven tested antibiotics (Table 2 and Table 3).

As shown in Figure 2, the isolation of MDR *E. coli* was higher in cloaca than chicken meat in both types of chicken.

### 2.3. Extended Spectrum Beta Lactamase (ESBL) Producing E. coli

Out of 212 identified *E. coli*, 10 (4.7%) isolates were screened and confirmed to be ESBL-producing *E. coli* (Table 4) All 10 isolates were found to be MDR. All (100%) ESBL producers were resistant to cefotaxime and ampicillin, 90% were resistant to tetracycline and trimethoprim-sulfamethoxazole, 40% were resistant to ciprofloxacin and 10% were resistant to imipenem. However, all 10 (100%) *E. coli* isolates were susceptible to gentamycin (Table 4). All confirmed ESBL-producing *E. coli* were isolated from one poultry slab at Stereo in Temeke district, and were mostly from spent layers.

### 2.4. Detection of CTX-M, TEM, SHV and PMQR Genes (qnrA, qnrB, qnrS and aac(6′)-lb-cr)

Plasmid mediated quinolones-resistant genes were detected in 8/10 ESBL-producing *E. coli* either as single genes or in combination with *CTX-M*, while TEM and SHV were not detected. The *qnrS* were present in eight (80%) of the isolates isolated from four spent layers’ meat, two spent layers’ cloaca, one broiler meat and one cloaca of broilers. All eight isolates with detected *qnrS* were resistant to ampicillin, sulfamethoxazole-trimethoprim and cefotaxime, seven of the eight were resistant to tetracycline, four of the eight were resistant to ciprofloxacin and three of the eight were resistant to imipenem. PMQR determinants *qnrA*, *qnrB* and *aac(6′)-lb-cr* were not detected in any of the *E. coli* isolates tested (Table 5, Figure 3 and Figure 4).

## 3. Discussion

In this study, isolation frequency of *E. coli* was more than half (55.2%), and most of the isolates (69.3%) were resistant to multiple drugs (MDR), some up to seven classes of antibiotics. The most frequent resistant combination was from tetracycline, ampicillin, sulfamethoxazole-trimethoprim and ciprofloxacin, which unfortunately are the most commonly used antibiotics in both humans and animals [10,11]. We found no significant differences in MDR *E. coli* between broilers and spent layers, even though broilers are raised in a relatively short period (four to six weeks) compared with spent layers (up to two years). This could be explained by the fact that antibiotics are used more intensely for growth promotion and prophylaxis in raising broilers than in spent layers [12]. For both types of chicken, cloaca had higher isolation frequency of MDR *E. coli* (25.2% to 32%) than in meat samples (20.4% to 22.4%), a trend that has also been observed in duck fecal samples [13], which indicates the epidemiological significance of chicken droppings in contaminating the environment, and acting as a potential driver of AMR spread [11,14]. We found significant difference in antimicrobial resistance rates between poultry slabs, indicating possible contribution of the slaughtering environment in contaminating poultry meat with MDR bacteria, which has been cited to be a factor [15,16,17]. However, in this study we did not investigate the sources of contamination.

For individual antibiotics, highest resistance was to tetracycline (91.9%), followed by sulfamethoxazole-trimethoprim (80.5%), ampicillin (70.9%) and ciprofloxacin (40.2%), a pattern that has also been reported previously [7,13,18]. These antibiotics are relatively cheap and are easily obtained over the counter [19,20,21,22], and often farmers do not follow withdrawal period [4,5].

On the other hand, cefotaxime, and especially imipenem, which are not commonly used [23], were less resistant. ESBL-producing *E. coli* were detected in 10/212 (4.7%) and quinolone resistance genes in 80% of them, supporting observations of several studies that have found a strong association between qnr-positive and ESBL-positive isolates [24,25,26,27]. Among the ESBL-producing, we only found CTX-M and not TEM or SHV, and unlike a study done in Niger that showed several qnr genes (*qnrA*, *qnrB* and *qnrS*) [28], we only found *qnrS*, and did not find *qnrA*, *qnrB* or *aac(6′)-lb-cr* in any of the ESBL isolates. Collectively, these findings suggest significant geographical differences in the occurrence of ESBL and quinolone resistance genes, supporting the need for their continuous surveillance in different settings.

The finding of quinolone resistance, which can rapidly spread along the food chain and in other ecosystems through plasmids [29,30,31], is significant, requiring very strict regulation in their use and, if possible, banning their use in animal food production. Tanzania has a number of acts and policies that are intended to control the quality of livestock production. Unfortunately, the Meat Industry Act of 2006 that gives a legal backing to support meat inspection to ensure quality does not explicitly address issues of drug residues in meat and meat products. Equally, the National Livestock Policy of 2006 and the National Agriculture Policy of 2013 do not address issues of antimicrobial use (AMU) and antimicrobial resistance (AMR) in livestock and agriculture sectors, respectively.

In all five slabs, we found improper handling of chicken carcasses and unregulated waste disposal from slaughter poultry slabs including blood, feces and wastewater disposed into municipal drains without either monitoring or treatment. Unfortunately, the National Environmental Policy of 1997 that is supposed to ensure food security through the promotion of production systems that are environmentally sound does not address the issues of environmental contamination with antimicrobials. Likewise, the Animal Diseases Act of 2003, which makes provisions for monitoring of production of animal products for disposal of animal carcasses, is silent on issues related to antimicrobials.

We strongly suggest the existing acts and policies, some of which are more than ten years old, be critically reviewed by stakeholders from human health, veterinary and environment sectors in order to curb AMU and AMR in livestock production and for protection of humans and the environment. The revised acts and policies should be reinforced through legislation. We also advocate for judicious use of antimicrobials in poultry, through improved hygiene, vaccinations and provision of extensive farmers’ education.

## 4. Materials and Methods

### 4.1. Study Area

The study was conducted in Dar es Salaam, the commercial city of Tanzania, which has a population of 4,364,541 people [3], with the highest production and consumption of chicken meat and eggs in Tanzania. The study involved five large poultry slabs in four Districts (Ilala, Ubungo, Temeke and Kinondoni). Approximately 20,000 chicken are slaughtered daily in these five poultry slaughter slabs, which provides about 80% of the chicken consumed in Dar es Salaam.

### 4.2. Study Design

This was a cross-sectional study conducted between January and June 2020 in four districts, which have the largest poultry slabs in Dar es Salaam. The slabs were Manzese, and Shekilango in the Ubungo district, Kisutu in the Ilala district, Mtambani in the Kinondoni district and Stereo in the Temeke district. In this study we targeted broilers and spent layers because they are raised intensively in overcrowded environments, and use of antimicrobials for prophylaxis, growth promotion and in management of infections is very high. Other types of poultry such as indigenous chickens were excluded from the study.

### 4.3. Sampling Technique

Using the purposive sampling technique, we selected 96 broilers and 96 spent layers, making a total of 192 chickens in all the five poultry slabs. Two samples (i.e., cloaca and meat swab) were collected from each chicken, making a total of 384 samples. Cloaca swabs were collected before chickens were slaughtered (at the entry point), while chicken meat swabs were collected after chickens were slaughtered (at the poultry slabs).

### 4.4. Specimen Collection

Chicken meat and cloaca swabs were collected aseptically using sterile cotton swabs and placed into a sterile tube containing 5 mL of Cary Blair transport medium (Oxoid, Basingstoke, UK). The collected samples were transported in a cool box at 2 to 8 °C containing a thermometer and were processed within 2 h of collection in the Microbiology Teaching Laboratory of the Muhimbili University and Allied Sciences (MUHAS).

### 4.5. Isolation and Identification of Enterobacteria

In the laboratory, swabs were inoculated onto the MacConkey agar (Oxoid, Basingstoke, UK) without antibiotics and incubated aerobically at 37 °C for 24 h. Identification of *E. coli* was done using colonial morphology, lactose fermentation and Gram stain. Lactose fermenters were subjected to conventional phenotypical identification using a set of biochemical tests including triple sugar iron agar (TSI), sulphur indole motility (SIM) agar and citrate utilization test. Confirmation was done using API 20E identification system for Enterobacteriaceae according to the instructions of the manufacturer (BioMérieux, Marcy-Etoile, France).

### 4.6. Screening and Confirmation of ESBL Production

Confirmed *E. coli* isolates were inoculated onto MacConkey agar containing 2 mg/L cefotaxime for preliminary screening of ESBL production. ESBL producers were confirmed using a combination disk method of cefotaxime 30 µg alone, combination with clavulanic acid (10 µg) and ceftazidime 30 µg alone and combination with clavulanic acid 10 µg. The difference of inhibition zone of more than or equal to 5 mm was confirmed as ESBL-positive [32]. *Klebsiella pneumoniae* ATCC 700603 was used as a positive control and *E. coli* ATCC 25922 as a negative strain.

### 4.7. Antimicrobial Susceptibility Testing

Antimicrobial susceptibility testing was done using the Kirby-Bauer disc diffusion method on Mueller Hinton Agar (Oxoid, Basingstoke, UK) based on CLSI 2018 guidelines [32]. Seven antibiotics from different classes were used, which included ampicillin (10 µg), tetracycline (30 µg), gentamycin (10 µg), ciprofloxacin (5 µg), imipenem (10 µg), sulfamethoxazole-trimethoprim (1.25/23.5 µg) and cefotaxime 30 µg [32].

Colonies of lactose fermenters identified as *E. coli* were emulsified into sterile saline to achieve turbidity equivalent to 0.5 McFarland standard, which is equivalent to 10^8^ cfu/mL [32]. Suspensions were spread onto Muller Hinton agar (MHA) using sterile cotton swabs and incubated aerobically at 37 °C for 16 to 18 h. The inhibition zone of each antibiotic was measured after 16 to 18 h of incubation, and results were interpreted according to the 2018 CLSI guidelines [32]. *E. coli* strain ATCC 29522 was used as a control strain. A strain was referred to be multidrug resistant (MDR) if it exhibited resistance to at least three different antibiotic classes [32].

### 4.8. Polymerase Chain Reaction (PCR)

#### 4.8.1. DNA Extraction

ESBL-producing *E. coli* isolates were inoculated on nutrient agar and incubated aerobically at 37 °C for 24 h. DNA was extracted by boiling in a water bath at 100 °C for 10 min, followed by centrifugation at 1500 rpm for 3 min. The supernatant containing DNA was transferred into sterile Eppendorf PCR tube (Eppendorf AG, Hamburg, Germany), and centrifugation and separation of supernatant were repeated three times. The concentration of DNA was determined by Nano drop spectrophotometer (Biochrom LTD, Cambridge, England) at 260/280 and 260/230 wavelength. DNA was stored at −20 °C, before being used for detection of ESBL genes (*CTX—M, TEM and SHV)* and PMQR genes (*qnrA*, *qnrB*, *qnrS* and *aac(6′)-Ib-cr*).

The Dream Tag DNA polymerase kit (Sigma-Aldrich Chemie GmbH, Taufkirchen, German) was used in detection of resistance genes. Total PCR reaction volumes were 25 µL, consisting of 10× dream Tag Buffer 5 µL, dNTP 2 mM 5 µL, forward and reverse primers were 1 µL each, DNA extract was 2 µL, Dream Tag DNA Polymerase (1.25 U) 1 µL and nuclease free water 10 µL. The primers used in amplification of respective *E. coli* resistance genes are listed in Table 6.

#### 4.8.2. Molecular Detection of CTX-M Genes

All ESBL-producing *E. coli* isolates were screened for the *CTX-M* gene using Uniplex PCR-based technique [33]. The process involved initial denaturation at 94 °C for 10 min, followed by 35 cycles of denaturation at 94 °C for 30 s, annealing at 58 °C for 30 s, extension at 72 °C for 60 s and final extension at 72 °C for 5 min [34].

#### 4.8.3. Detection of TEM and SHV Genes

Extended spectrum beta lactamase *TEM* and SHV genes were screened by uniplex PCR-based assay [37] with the following amplification conditions: initial denaturation at 96 °C for 5 min, followed by 35 cycles of denaturation at 96 °C for 1 min, annealing at 58 °C (TEM) and at 60 °C (SHV) for 1 min, extension at 72 °C for 1 min and final extension at 72 °C for 10 min [37].

#### 4.8.4. Detection of PMQR Genes (*qnrA*, *qnrB* and *qnrS*)

PMQR-resistant genes (*qnrA*, *qnrB* and *qnrS*) were amplified and detected using multiplex PCR assay [14]. The process involved initial denaturation at 94 °C for 5 min, followed by 32 cycles of denaturation at 94 °C for 45 s, annealing at 53 °C for 1 min, extension at 72 °C for 1 min and final extension at 72 °C for 10 min [14].

#### 4.8.5. Detection of *aac(6′)-lb-cr* Gene

*aac(6′)–lb-cr* genes were screened by uniplex PCR-based assay [35] using the following amplification conditions: initial denaturation at 94 °C for 5 min, followed by 34 cycles of denaturation at 94 °C for 45 s, annealing at 55 °C for 45 s, extension at 72 °C for 45 s and final extension at 72 °C for 10 min [36].

### 4.9. Data Analysis

The data were entered into Microsoft Excel; proportions were analyzed by Chi-square test. A paired *t*-test assuming unequal variance was used for comparing overall prevalence and comparing resistance rate among tested antibiotics in SPSS version 16 software. A *p*-value (<0.05) was considered to be statistically significant.

## 5. Conclusions

The high levels of resistance to antibiotics seen in this study has several implications: (i) there is over use of antibiotics in poultry production; (ii) these agents are ineffective in either prophylaxis or treatement of infections in poultry farming; (iii) there is a serious public health threat (through antimicrobial residues in meat); and (iv) increased possibility of development and spread of antimicrobial resistance in the environment. Therefore, urgent measures are required to reduce the use of antibiotics in poultry production at the farm level and to improve hygiene practices at poultry slaughter. In addition, the present acts and policies governing animal food production should be revised to provide legislation to enforce judicious use of antimicrobial agents.

## Figures and Tables

**Figure 1 antibiotics-10-00343-f001:**
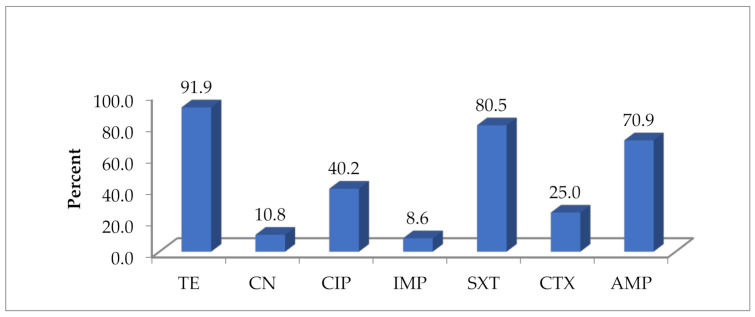
Overall antibiotic resistance pattern of *E. coli* isolates in five poultry slabs (*n* = 212). TE = tetracycline, CN = gentamycin, CIP = ciprofloxacin, IMP = imipenem, SXT = trimethoprim-sulfamethoxazole, CTX = cefotaxime, AMP = ampicillin.

**Figure 2 antibiotics-10-00343-f002:**
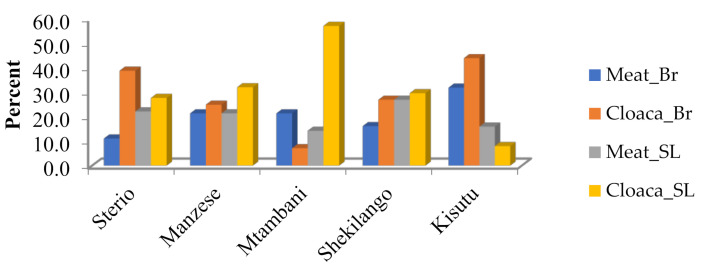
Multiple drug resistance of *E. coli* by location of poultry slab.

**Figure 3 antibiotics-10-00343-f003:**
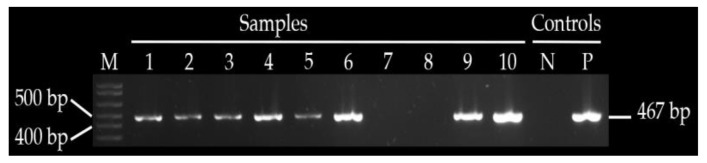
Shows amplified *qnrS* gene in sample 1–6, 9 and 10, M—1 kb ladder, NC—negative control, PC—positive control.

**Figure 4 antibiotics-10-00343-f004:**
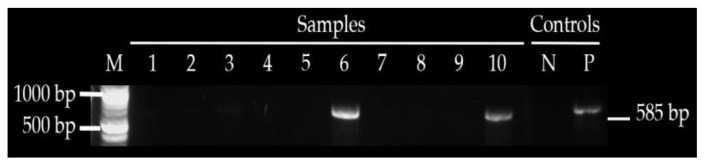
Shows amplified *CTX—M* in sample 6 and 10, M—1 kb ladder, NC—negative control and PC—positive control.

**Table 1 antibiotics-10-00343-t001:** Frequency of MDR and non-MDR *Escherichia coli* isolated from the selected poultry slabs in Dar es Salaam (*n* = 384).

Poultry Slabs	MDR		Not MDR	
	*n*	%	*n*	%
Stereo	18	94.7	1	5.3
Manzese	28	70.0	12	30.0
Mtambani	14	70.0	6	30.0
Shekilango	37	86.0	6	14.0
Kisutu	50	55.6	40	44.4
Total	147	69.3	65	30.7

MDR—multidrug resistant.

**Table 2 antibiotics-10-00343-t002:** Classes of antimicrobial patterns resisted *n* (%).

MDR *E. coli* Isolates	Classes of Antibiotics				
	3	4	5	6	7
147	72(49)	49(33.3)	21(14.3)	4(2.7)	1(0.7)

**Table 3 antibiotics-10-00343-t003:** Antimicrobial resistance pattern of multidrug-resistant *E. coli.*

Antibiotic Combination	Number of Isolates	%	Number of Antibiotics Classes
TE, CN, CIP	1	0.7	3
TE, CN, SXT	1	0.7	3
TE, CIP, SXT	17	11.6	3
TE, CN, AMP	1	0.7	3
TE, IMP, SXT	3	2.0	3
TE, CIP, AMP	8	5.4	3
TE, CN, CIP, SXT	3	2.0	4
TE, IMP, AMP	1	0.7	3
TE, SXT, AMP	39	26.5	3
TE, CIP, IMP, SXT	1	0.7	4
CIP, SXT, AMP	1	0.7	3
TE, CN, SXT, AMP	4	2.7	4
TE, CIP, IMP, AMP	1	0.7	4
TE, CIP, SXT, AMP	29	19.7	4
TE, IMP, SXT, AMP	1	0.7	4
TE, CN, CIP, IMP, AMP	1	0.7	5
TE, IMP, CTX, AMP	1	0.7	4
TE, CN, CIP, SXT, AMP	4	2.7	5
TE, SXT, CTX, AMP	9	6.1	4
TE, CIP, IMP, SXT, AMP	3	2.0	5
TE, CN, SXT, CTX, AMP	3	2.0	5
TE, CIP, SXT, CTX, AMP	9	6.1	5
TE, CN, CIP, IMP, SXT, AMP	2	1.4	6
TE, IMP, SXT, CTX, AMP	1	0.7	5
TE, CN, CIP, SXT, CTX, AMP	2	1.4	6
TE, CN, CIP, IMP, SXT, CTX, AMP	1	0.7	7

TE = tetracycline, AMP = ampicillin, SXT = trimethoprim-sulfamethoxazole, CIP = ciprofloxacin, CTX = cefotaxime, CN = gentamycin, IMP = imipenem.

**Table 4 antibiotics-10-00343-t004:** Antimicrobial resistance of ESBL-producing *E. coli* isolates (*n* = 10).

				Chicken Category			
Antibiotic	Isolates (*n*)	%	Meat SL	Cloaca SL	Meat Br	Cloaca Br	Total
TE	9/10	90	3	4	1	1	**9**
CN	0/10	0	0	0	0	0	**0**
CIP	4/10	40	3	1	0	0	**4**
IMP	1/10	10	1	0	0	0	**1**
SXT	9/10	90	3	4	1	1	**9**
CTX	10/10	100	4	4	1	1	**10**
AMP	10/10	100	4	4	1	1	**10**

Meat SL = spent layers’ meat, cloaca SL = spent layers’ cloaca, cloaca meat Br = broiler meat, Br = broiler cloaca, TE = tetracycline, CN = gentamycin, CIP = ciprofloxacin, IPM = imipenem, SXT = trimethoprim-sulfamethoxazole, CTX = cefotaxime, AMP = ampicillin.

**Table 5 antibiotics-10-00343-t005:** Distribution of ESBL- and PMQR-encoding genes by PCR (*n* = 10).

Detected Genes	*E. coli* No (%)	Spent Layers Meat	Spent Layers Cloaca	Broiler Meat	Broiler Cloaca
*CTX-M*	2/10 (20)	1	1	0	0
*TEM*	0/10 (0.0)	0	0	0	0
*SHV*	0/10 (0.0)	0	0	0	0
*QnrA*	0/10 (0.0)	0	0	0	0
*QnrB*	0/10 (0.0)	0	0	0	0
*QnrS*	8/10 (80)	4	2	1	1
*aac(6′)-Ib-cr*	0/10 (0.0)	0	0	0	0
Total	10/10(100)	5	3	1	1

**Table 6 antibiotics-10-00343-t006:** List of primers used.

Gene	Primer Set	Amplicon Size	Reference
*CTX*-*M*	F: SCSATGTGCAGYACCAGTAA R: ACCAGAAYVAGCGGBGC	585 bp	[33,34]
*QnrA*	F: TCAGCAAGAGGATTTCTCA R: GGCAGCACTATTACTCCCA	627 bp	[35]
*QnrB*	F: GGMATHGAAATTCGCCACTG R: TTTGCYGYYCGCCAGTCGAA	264 bp	[35]
*QnrS*	F: ATGGAAACCTACAATCATAC R: AAAAACACCTCGACTTAAGT	467 bp	[35]
*aac(6′)-Ib-cr*	F: TTGCGATGCTCTATGAGTGGCTA R: CTCGAATGCCTGGCGTGTTT	482 bp	[34,36]
TEM	F: ATGAGTATTCAACATTTCCG R: CTGACAGTTACCAATGCTTA	867 bp	[37]
SHV	F: GGTTATGCGTTATATTCGCC R: TTAGCGTTGCCAGTGCTC	867 bp	[37]

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
