# Peer review of "Occurrence of Multidrug Resistant Escherichia coli in Raw Meat and Cloaca Swabs in Poultry Processed in Slaughter Slabs in Dar es Salaam, Tanzania"

_antibiotics, 2021, doi:10.3390/antibiotics10040343_

Round 1
Reviewer 1 Report
The manuscript has presented the occurrence of multidrug resistance E. coli from raw meat and cloaca samples by PCR methods through determining resistant genes. The manuscript was well written and explanations. However, the authors are suggested proof-reading the manuscript carefully to avoid some minor mistakes before publications since there were a lot of unnecessary mistakes along the manuscript.
- Authors name: the "*" indicated the correspondence should be placed after affiliation ( Fauster X. Mgaya 1,2,*).
- There are a lot of typo mistakes along with the manuscript such as oC, space between number and unit (30µL, 5mm,...), etc... The authors are suggested to check and correct all.
- There were many table of data which could be made the reader feel boring with less-eye catching. Therefore, some data can be transfer into graph format which can be improved the interest for the reader.
- The writing style should be identical for common expression (16 to 18-hours VS 16 to 18 hours). This writing style should be checked and corrected.
- Four sections from 4.8 to 4.11 can be combined with all the PCR's related information and re-numbered as 4.9 Polymerase Chain Reaction (PCR) for easier understanding.
- References were some mistakes such as ref. 14 (space between "." and "Int", ref. 17 lack of page numbers. Also, the style of references also not identical (Ann N Y Acad Sci. 2019; 1441(1):17-30.; BMC Microbiol. 2018; 18, 174-181; Antimicrob Agents Chemother. 2006; 50 (4):1178–82; Livestock Rural Res Dev. 2012; 24(10):1–14). They should be checked and corrected for all those confusing styles.
Author Response
Responses for the reviewers are in the attached word files

Reviewer 2 Report
Antibiotic resistance is a growing global menace. In this study, Mgaya et al. determine the occurrence of multidrug-resistant E. coli isolated from raw chicken meat and cloaca. This article will be of broad interest to readers of the antibiotics journal. However, some items need to be addressed before publication:
- My primary concern is the molecular detection of CTX-M and PMQR genes. Several bands in Figure 1 are over-saturated. Meanwhile, it is very hard to see any band in Figure 2, even in the positive control lane. Authors are suggested to optimize the condition and re-run the DNA gels.
- The blaCTX-M is now the predominant and the most extensively disseminated ESBL genotypes found among E. coli isolates. However, authors should include the detection of blaTEM, and blaSHV since only 20% ESBL producing coli isolates carries CTX-M gene in this study.
- It is unclear why authors use both 2018 and 2019 CLSI guidelines on Page 7. Authors should clarify it.
- It would be nice if the authors can correct the format of the third paragraph on Page 6.
- Following the instruction to author file, authors are suggested to move 4.13. to Institutional Review Board Statement between Author Contributions and Conflicts of Interest.
Author Response
I have replied to the reviewer 2 comments,
Find the attached word documents

Round 2
Reviewer 2 Report
The authors have addressed my concerns. The revised manuscript looks good.